# Inverse Problem Algorithm-Based Time-Resolved Imaging of Head and Neck Computed Tomography Angiography Contrast Kinetics with Clinical Testification

**DOI:** 10.3390/diagnostics13213354

**Published:** 2023-10-31

**Authors:** Chih-Sheng Lin, Bing-Ru Peng, Hong-Bing Ma, Ke-Lin Chen, Tsung-Han Lin, Lung-Kwang Pan, Ya-Hui Lin

**Affiliations:** 1Department of Radiology, BenQ Medical Center, Affiliated BenQ Hospital of Nanjing Medical University, Nanjing 211166, China; chihsheng.lin@benqmedicalcenter.com (C.-S.L.); david.ma@benqmedicalcenter.com (H.-B.M.); 2Department of Medical Imaging and Radiological Science, Central Taiwan University of Science and Technology, Takun, Taichung 406, Taiwan; a0953939793@gmail.com (B.-R.P.); linv926@gmail.com (T.-H.L.); lkpan@ctust.edu.tw (L.-K.P.); 3Department of Radiology, First Affiliated Hospital of Ningbo University, Ningbo 315012, China; chenkl2m@me.com; 4Department of Traditional Chinese Medicine, Taichung Armed Forces General Hospital, Taichung 411, Taiwan; 5Department of Clinical Pharmacy, Taichung Armed Forces General Hospital, Taichung 411, Taiwan

**Keywords:** head and neck CT angiography, inverse problem algorithm (IPA), imaging quality, semi-empirical formula, *STATISTICA 7.0*, ANOVA

## Abstract

This study mitigated the challenge of head and neck CT angiography by IPA-based time-resolved imaging of contrast kinetics. To this end, 627 cerebral hemorrhage patients with dizziness, brain aneurysm, stroke, or hemorrhagic stroke diagnosis were randomly categorized into three groups, namely, the original dataset (450), verification group (112), and in vivo testified group (65), in the Affiliated BenQ Hospital of Nanjing Medical University. In the first stage, seven risk factors were assigned: age, CTA tube voltage, body surface area, heart rate per minute, cardiac output blood per minute, the actual injected amount of contrast media, and CTA delayed trigger timing. The expectation value of the semi-empirical formula was the CTA number of the patient’s left artery (LA). Accordingly, 29 items of the first-order nonlinear equation were calculated via the inverse problem analysis (IPA) technique run in the *STATISTICA 7.0* program, yielding a loss function and variance of 3.1837 and 0.8892, respectively. A dimensionless AT was proposed to imply the coincidence, with a lower AT indicating a smaller deviation between theoretical and practical values. The derived formula was confirmed for the verification group of 112 patients, reaching high coincidence, with average AT_avg_ and standard deviation values of 3.57% and 3.06%, respectively. In the second stage, the formula was refined to find the optimal amount of contrast media for the CTA number of LA approaching 400. Finally, the above procedure was applied to head and neck CTA images of the third group of 65 patients, reaching an average CTA number of LA of 407.8 ± 16.2 and finding no significant fluctuations.

## 1. Introduction

Time-resolved imaging of contrast kinetics for head and neck computed tomography angiography (H&N CTA) via the inverse problem algorithm (IPA) technique was investigated in this study. Producing optimal CTA images has been a challenge in clinical diagnosis since many factors of the CTA facility must be inclusively optimized and compiled with the physical condition or metabolism of various patients to obtain CTA images meeting the diagnostic standards.

Several researchers have tried to mitigate this problem by optimizing the CTA facility parameters for head and neck imaging. Ma et al. [1] tried to adopt a low tube voltage to acquire better head and neck CTA images for the two groups under study. Alternatively, Wang et al. [2], Benz et al. [3], and Saade et al. [4] applied low-dose contrast media to improve imaging quality for two patient groups. Xia et al. [5] and Zhang et al. [6] reported that both low-tube voltage and low-contrast medium effectively improved the H&N CTA imaging quality. The analysis of 37 studies collected from 2011 to 2015 in the survey of Shen et al. [7] confirmed the conclusion that the “double low approach” (i.e., low tube voltage and low-dose contrast media) was a good solution that reduced the radiation dose and still maintained good imaging quality in H&N CTA. Based on a study of 600 patients undergoing H&N CTA examination, Lin et al. [8] reported that the above imaging quality was dominated by either CTA tube voltage or concentration of injected contrast media, with minor contributions of other factors (injection into different hands, field of view, flow rate of contrast media, or rotation time for one CTA loop), which failed to pass the ANOVA test of significance.

However, a good imaging quality of H&N CTA also needs an integrated consideration of either the CTA facility or the personal characteristics of each patient altogether. To this end, Liang et al. [9] compiled six factors, including patients’ biological indices (age, mean arterial pressure (MAP), heart rate, and body surface area (BSA)) or CTA facility parameters (concentration of contrast media and injection pressure), to compose a semi-empirical formula by the IPA technique. They predicted the optimal trigger timing of H&N CTA, reaching an 89% accuracy for 802 patients under study, with further verification via 199 actual patients in clinical examination, proving the feasibility of the IPA technique in resolving this medical issue. Thus, a preliminary survey collecting several essential parameters of patients is required to optimize the CT imaging quality instead of cumbersome and/or time-consuming examinations in the clinical field.

Accordingly, this study also applied IPA to optimize the suitable amount of contrast media, splitting it into two stages. The first stage assigned seven factors (age, BSA, heart rate, CTA tube voltage, cardiac output blood per minute, actual amount of contrast media, and CTA scan delay trigger timing) to run a program with a customized first-order nonlinear semi-empirical formula according to the IPA algorithm definition. The expectation value of the IPA technique was preset as the CT scanned number of the patient’s left artery (LA). In total, 29 coefficients of the nonlinear semi-empirical formula were computed by *the STATISTICA 7.0* program [10]. The second stage derived the expected amount of contrast media by replacing the original with maximal or minimal amounts in the semi-empirical formula to obtain the estimated amount of practical contrast media for individual patients from a correlated binominal equation. In addition, the derived data were applied to 65 patients in the clinical survey and successfully reached the demanded CT scan number of the patient’s left artery (LA). The definition of the seven adopted factors, IPA technique, interpretation of the derived coefficients in the semi-empirical formula, calculated results, and practical verification are discussed in the next sections. The details of the IPA realization via the default program *STATISTICA 7.0* are given in the Appendix A.

## 2. Methodology

### 2.1. Stage 1

#### 2.1.1. Basics of the Inverse Problem Algorithm

Consider the first-order linear equation y=xβ, where *y* is the expected value, while the sensitivity of *x* to *y* is reflected by β=y/x. Assume that *y* = *y* [450 × 1] is the expected value (also referred to as the ideal amount of contrast media), correlating 29-term coefficients *M* [29 × 1]. Then, the respective correlation equation can be expressed as follows:(1)Y=VM
(2)$$\left|\begin{array}{c} y_{1}\\ y_{2}\\ y_{3}\\ \vdots \\ y_{n} \end{array}\right|=\left|\begin{array}{ccccc} v_{11} & v_{12} & . & . & v_{1m}\\ v_{21} & v_{22} & . & . & v_{2m}\\ v_{31} & v_{32} & . & . & v_{3m}\\ \vdots & \vdots & \vdots & \vdots & \vdots \\ v_{n1} & v_{n2} & . & . & v_{nm} \end{array}\right|\left|\begin{array}{c} M_{1}\\ M_{2}\\ M_{3}\\ \vdots \\ M_{m} \end{array}\right|$$

If ∅ is the standard loss function and *n* is the number of patients, then
(3)$${\emptyset =\left\|VM-Y\right\|}_{2}^{2}\;\;$$
(4)$$\nabla_{M}\emptyset =2\left(V^{T}\cdot VM-V^{T}Y\right)=0$$
(5)$$V^{T}\cdot VM=V^{T}Y$$
(6)$$M={\left(V^{T}\cdot V\right)}^{-1}\cdot V^{T}\cdot Y$$
where *V* and *V^T^* represent the direct and transpose dataset matrices, respectively, of the risk factors and their bivariate cross-interactions [450 × 29]. The inverse matrix (*V^T^*·*V*) in Equation (6) allows one to construct the column matrix *M* with 29 coefficients of the final semi-empirical equation [11]. The above formulas are incorporated into *STATISTICA 7.0*, which searches for the minimal loss function Φ. The obtained compromised solution can be further customized for the user’s specific demands.

Using the representative number of patient biological indices and/or CTA facility parameters, the quantitative expectation of a particular syndrome is computed via IPA, ensuring solution convergence when constructing the inverse matrix of biological datasets. Eventually, it yields a 29-term semi-empirical formula that can easily furnish a supplementary recommendation on clinical diagnosis based on scanned images of CT angiography, sonography, cardiac X-ray, or other radiological facilities [11].

#### 2.1.2. IPA Derivation of the 29-Term Formula

The IPA approach considers only single-factor contributions and two-factor cross-interactions, neglecting triple (e.g., *v*1 *× v*2 *× v*3) and quadruple (e.g., *v*1 *× v*2 *× v*3 *× v*4) contributions. It ensures the numerical solution’s convergence by merging multiple residual cross-interactions into the final constant term, treated as a minor oscillation [12]. The following mathematical expression is used:(7)$$\begin{array}{cc} v8= & a1\times v1+a2\times v2+a3\times v3+a4\times v4+a5\times v5+a6\times v6+a7\times v7+a8\times v1\times v2\\ & +a9\times v1\times v3+a10\times v1\times v4+a11\times v1\times v5+a12\times v1\times v6+a13\times v1\times v7\\ & +a14\times v2\times v3+a15\times v2\times v4+a16\times v2\times v5+a17\times v2\times v6+a18\times v2\times v7\\ & +a19\times v3\times v4+a20\times v3\times v5+a21\times v3\times v6+a22\times v3\times v7+a23\times v4\times v5\\ & +a24\times v4\times v6+a25\times v4\times v7+a26\times v5\times v6+a27\times v5\times v7+a28\times v6\times v7\\ & +a29\; \end{array}$$

Here, *v*8 is the expectation value (i.e., the CTA number of LA) derived via seven risk factors treated as variables (*v*1~*v*7).

#### 2.1.3. CTA Number of LA and Seven Risk Factors

The CTA number of the left artery (LA) for each patient under CTA examination was recommended to be controlled within a 350–450 range for optimal imaging quality [13,14,15]. Thus, according to the clinical radiologist’s suggestion, the expected value of the CTA number for the patient’s LA was preset at 400.

Seven essential biological indices or characteristics of the CTA facility were adopted as risk factors and listed as (1) age (yr); (2) CTA tube voltage (kVp, V); (3) BSA (m^2^); (4) heart rate per minute (HR, #/min); (5) cardiac output blood per minute (CO, L/min); (6) actual injected amount of contrast media in c.c. (CM, c.c.); and (7) CTA delayed trigger timing (DTT, s). The body surface area (BSA) strongly correlates with human metabolic mechanisms and is defined as $\sqrt{H\times W/3600}$) [m^2^], (H: height [cm], W: weight [kg]) [16,17]. Four of the seven assigned risk factors were personal biological indices of the patient (age, BSA, HR, and CO), whereas the remaining three were CTA facility characteristics (kVp, CM, and DTT). Thus, the risk factors covered both patient data and CTA facility operational parameters, in contrast to earlier studies that focused either on the former or latter parameters.

Each risk factor’s dimensionality should be unified before the IPA processing run in the *STATISTICA 7.0* program. To this end, their standard normalization into the domain range [−1; +1] was performed for each risk factor’s reading X^*^ as follows:(8)$$X^{*}=\frac{X-\frac{X_{max}+X_{min}}{2}}{\frac{X_{max}-X_{min}}{2}}$$
where *X*, *X_min_*, and *X_max_* are the respective risk factor’s original, minimal, and maximal readings (*v*_1_–*v*_7_). Given the maximal and minimal age readings *v*_1_ (90 and 24 yrs), the original ages (60 and 33 yrs) of patients No. 195 and No. 274 were normalized to +0.0909 and −0.7273. Thus, the normalized age scale ranged from −1.0 to +1.0.

Readings of the seven factors and their actual CTA numbers of LA (listed in Table 1) were obtained for 450 cerebral hemorrhage patients with dizziness, brain aneurysm, stroke, or hemorrhagic stroke diagnosis randomly selected from the dataset of 562 such patients reported in the Affiliated BenQ Hospital of the Nanjing Medical University from 1 January 2020 to 30 June 2023. The remaining 112 patients with the same syndrome were included in the verification group.

### 2.2. Stage 2

#### Extreme Values of *CM_max_* and *CM_min_*

Factor six (*v*6) in Equation (7) was defined as the actual injected amount of CM in c.c. Replacing *v*6 with *v_max_*(+1) or *v_min_*(−1) in Equation (7), we obtain the *CM_max_* or *CM_min_* expectations. Then, the binominal equation takes the following form [18]:(9)$$\frac{CM_{\mathrm{expect}}-CM_{min}}{CM_{max}-CM_{min}}=\frac{v8(\mathrm{expect})-v8\left(v6_{min}\right)}{v8\left(v6_{max}\right)-v8\left(v6_{min}\right)}$$
*CM_max_* and *CM_min_* are the maximal or minimal dosages of injected contrast media as calculated from the derived semi-empirical formula, and *CM_expect_* is the theoretical amount c.c of CM that should be injected into the specific patient. In contrast, *v*8(*v*6*_max_*) or *v*8(*v*6*_min_*) is the normalized maximal (+1.0) or minimal (−1.0) CTA number of the LA for the specific patient who was assumed to inject either the maximal or minimal amount of CM. While this is a theoretical presumption in this study, the linearity of the correlation between *v*6 (CM) and *v*8 (the CTA number of LA), as defined in Equation (9), can be sustained in this finite realm. Thus, the exact amount of CM can be easily derived once *v*8 (*expect*) is forced to equal 400, i.e., the optimal CTA number of LA in this calculation.

For instance, patient No. 1 in the testified group (with 65 cases adopted in this study) has *CM_max_* = 75 c.c. and *CM_min_* = 17 c.c. (cf. Table 1), with the derived *v*8(*v*6*_max_*) = 552 and *v*8(*v*6*_min_*) = 252. Given that this derivation represents the theoretical estimation of the CTA number of LA for the specific patient from the semi-empirical formula, the optimal amount c.c. of CM injection to this patient should be 45.6 c.c. to achieve a 400 CTA number of LA as listed below:(10)$$\frac{{CM}_{expect}-17}{75-17}=\frac{400-252}{552-252};\;\;thus,\;{CM}_{expect}=45.6\;c.c.$$

Parameter *v*8 is each patient’s expected CTA number of LA, so it should be calculated according to specific biological indices of each patient as risk factors in the *STATISTICA 7.0* program (see Appendix A for details). Thus, Equation (9) needed customization for each patient to obtain optimal amounts of injected CM.

## 3. Results

### 3.1. Raw Data Normalization

The seven risk factors under study were normalized and summarized in Table 2. If the patient’s group biological indices or/and facility characteristics are normally distributed within the [−1.0; +1.0] range, their mean values should be close to zero. As seen in Table 2, only age’s mean value (−0.02) satisfied this requirement, in contrast to those of the remaining six factors, ranging from −0.42 to −0.13, thus not complying with the normal distribution of the statistical data from 450 patients.

### 3.2. Calculation Procedure and Program Performance of the Customized IPA Algorithm

A spreadsheet of *STATISTICA 7.0* in Figure 1 illustrates the model, variables, loss function, and its derived value, variance, and correlation coefficient *R*.

As seen in Figure 1, the derived customized loss function (Φ = [OBS-PRED]^2^) was 3.184, while its zero value would indicate a 100% coincidence between the predicted and actual results. Other indices (the sample variance of 0.89 and regression correlation coefficient of 0.943) match the derived prediction well with the actual data matrix. The 29 coefficients of the semi-empirical formula (7) were derived and are listed in Table 3. Given the performed normalization of all risk factors from −1 to +1, those with high coefficients in Table 3 mostly contribute to the results and, thus, are the most influencing factors controlling the CTA number of LA.

## 4. Discussion

### 4.1. Verifying the Predicted CTA Number of LA by the Derived Semi-Empirical Formula

Another group of 112 patients randomly selected from the original patient group in this study was assigned as the verification group to verify the prediction reliability of the CTA number of LA via the derived semi-empirical formula. In doing so, the biological indices of the verification group were inserted into the semi-empirical formula and then processed to obtain the predicted CTA number of LA. Table 4 lists the detailed information on the verification group. As demonstrated, each risk factor’s maximal or minimum value falls into a similar range as the original group. Figure 2 shows that the deviation of a verification group of 112 patients’ data coincided with the original data group of 450 patients. As depicted, two data groups were merged consistently along the axis of the actual CTA number of LA. In addition, the X- or Y-axis was preset as the CTA number of LA divided by 400 since 400 was the preferable CTA number for cross-comparison. Specifically, agreement (AT) was defined as the difference between the actual and predicted values divided by the actual value of the CTA number of LA [19]. Therefore, the average AT_avg_ and standard deviation of the 112 individual ATs were 3.57% and 3.06%, respectively, implying close agreement between the actual and predicted values [20,21]. Figure 3 illustrates the distribution of 112 individual ATs in this study. As demonstrated, 42 out of 112 ATs are below 2%, indicating a reliable prediction of the CTA number of LA [11,12].

### 4.2. Interpreting the Dominant Factors Controlling the Prediction of the CTA Number of LA

Five of seven factors provided dominant contributions to the prediction of the CTA number of LA for individual patients, according to the performed analysis of derived coefficients of each factor and their cross-interaction (cf. Table 3). Notably, only tube voltage had a dominant contribution if no cross-interactions among factors were taken into account. However, cross-interaction among factors can provide additional dimensionality in the IPA-based analysis since any vector A×B can create another vector of (A × B) that is vertical to either the A or B vector for adding one more degree of freedom (DoF) and increasing the convergence of the semi-empirical formula [11,12]. Thus, the contribution from cross-interaction between two factors is also important in IPA application.

The other four risk factors (BSA, CO, CM, and DTT) showed dominance in affecting cross-interactions, whereas age and HR exhibited no significance, according to *STATISTICA 7.0* results. Possibly, the age or HR-related human metabolic mechanism is too complicated to grasp their importance by this model and requires more detailed analysis, so they cannot be removed from the list of risk factors to avoid misinterpretation of contributions and possible correlations among factors.

Specifically, Lin et al. [8] claimed that the Taguchi optimization analysis proved that tube voltage was less important than tube current. At the same time, the concentration of contrast media (CM) provided a dominant contribution to the CTA imaging quality of 600 patients. Liang et al. [9] confirmed that age, BSA, and MAP were three dominant factors in CTA imaging quality assurance. This study revealed that BSA and CM provided a dominant contribution to this task. Some researchers [2,3,4,5,6] also reduced the contrast media dosage to gain high CTA imaging quality, but their results were limited to comparing the experimental and control groups. In contrast, the refined IPA technique applied in this study reliably assessed the contribution of each risk factor with an account of their cross-interactions, checked their significance, and ranked their importance.

### 4.3. In Vivo Testification of 65 Patients

Sixty-five patients with similar syndromes were randomly adopted as the testified group in the follow-up study. All patient biological indices were normalized, and then the semi-empirical Equation (7) with risk factor No. 6 (*CM*) was substituted by either *CM_max_* or *CM_min_* and then inserted into Equation (8) to obtain *CM_expect_* (25.9~77.7 c.c.). Figure 4 depicts the actual CTA numbers of LA for the sixty-five patients obtained via Equation (9). When the optimal CTA number was preset to 400, the scanned data in the CTA examination slightly fluctuated near 400. The exact average and standard deviations were 407.8 ± 16.2, indicating no significant fluctuation.

Figure 5 shows the CT images obtained with optimal, too-low, and too-high CTA numbers of LA for six of the sixty-five patients in the testified group. The optimal imaging quality provided a suitable diagnosis and reduced any misreading or misinterpretation of imaging in clinical treatment.

## 5. Conclusions

This study mitigated the head and neck CT angiography challenge by IPA-based time-resolved imaging of contrast kinetics. Six hundred twenty-seven cerebral hemorrhage patients with dizziness, brain aneurysm, stroke, or hemorrhagic stroke diagnosis were randomly categorized into three groups, namely, the original dataset (450), verification group (112), and in vivo testified group (65), at the Affiliated BenQ Hospital of Nanjing Medical University. In the first stage, a 29-item first-order nonlinear semi-empirical formula was established using seven risk factors for each patient in the original dataset group. The default numerical program run in *STATISTICA 7.0* compiled all data, providing a compromised solution via the IPA technique. The derived semi-empirical formula was then applied to the verification group of 112 patients, reaching a high coincidence, with an average AT_avg_ and standard deviation of 3.57% and 3.06%, respectively. In the second stage, the optimal amount of contrast media was adjusted to reach the CTA number of LA equal to 400. The average CTA number in the testified group of 65 patients was 407.8 ± 16.2, showing no significant fluctuations. The IPA algorithm was successfully applied to the preliminary survey of CT diagnosis. It reliably assessed the contribution of each risk factor with an account of their cross-interactions, checked their significance, and ranked their importance. Using the essential parameters of individual patients, the amount of contrast media needed to reach the optimal CT number of LA was predicted, which has practical importance in clinical examination.

## Figures and Tables

**Figure 1 diagnostics-13-03354-f001:**
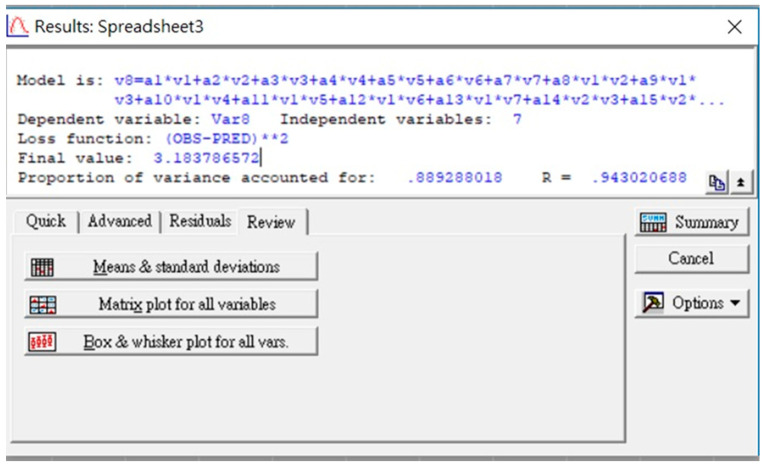
The *STATISTICA 7.0* spreadsheet with formulas and computation results.

**Figure 2 diagnostics-13-03354-f002:**
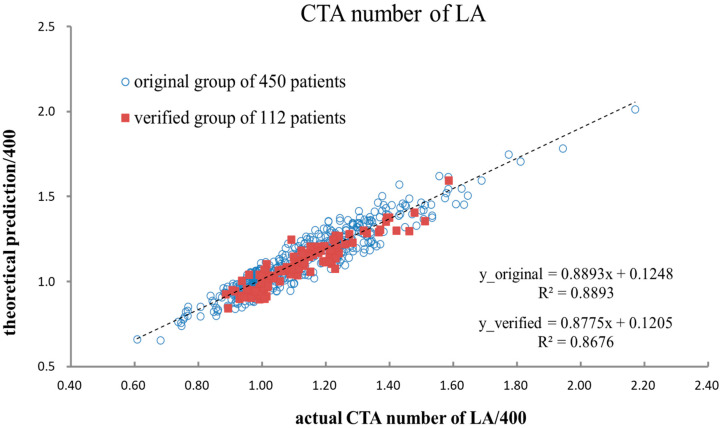
The actual and predicted CTA numbers of LA for the original 450 patients and 112 verification group patients, according to *STATISTICA 7.0*-derived linear regression.

**Figure 3 diagnostics-13-03354-f003:**
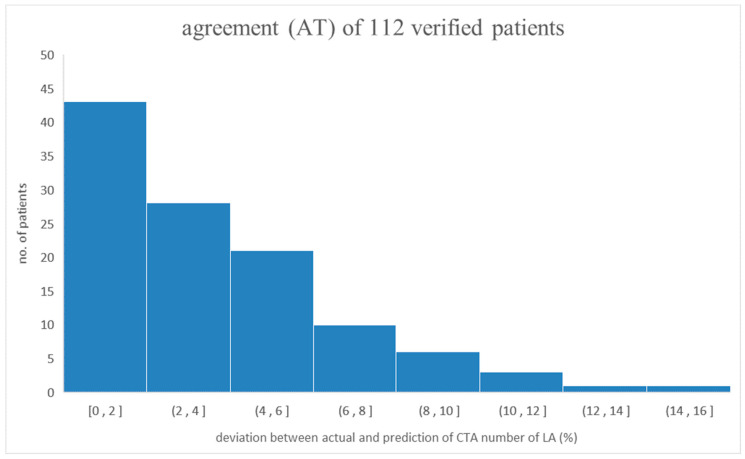
The distribution of 112 individual ATs in this study. As demonstrated, most ATs lie below 6%, showing the reliability of the IPA-based prediction of patients’ CTA numbers of LA.

**Figure 4 diagnostics-13-03354-f004:**
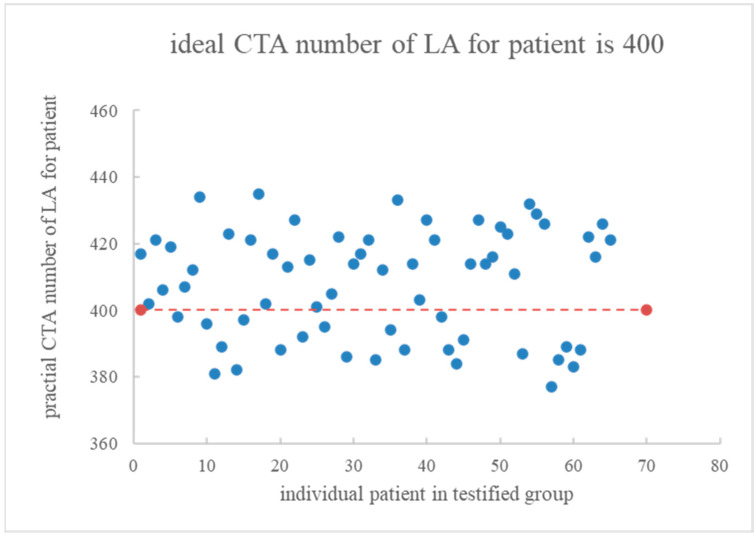
Each actual CTA number of LA was recorded for 65 patients in the testified group. The optimal CTA number was preset at 400.

**Figure 5 diagnostics-13-03354-f005:**
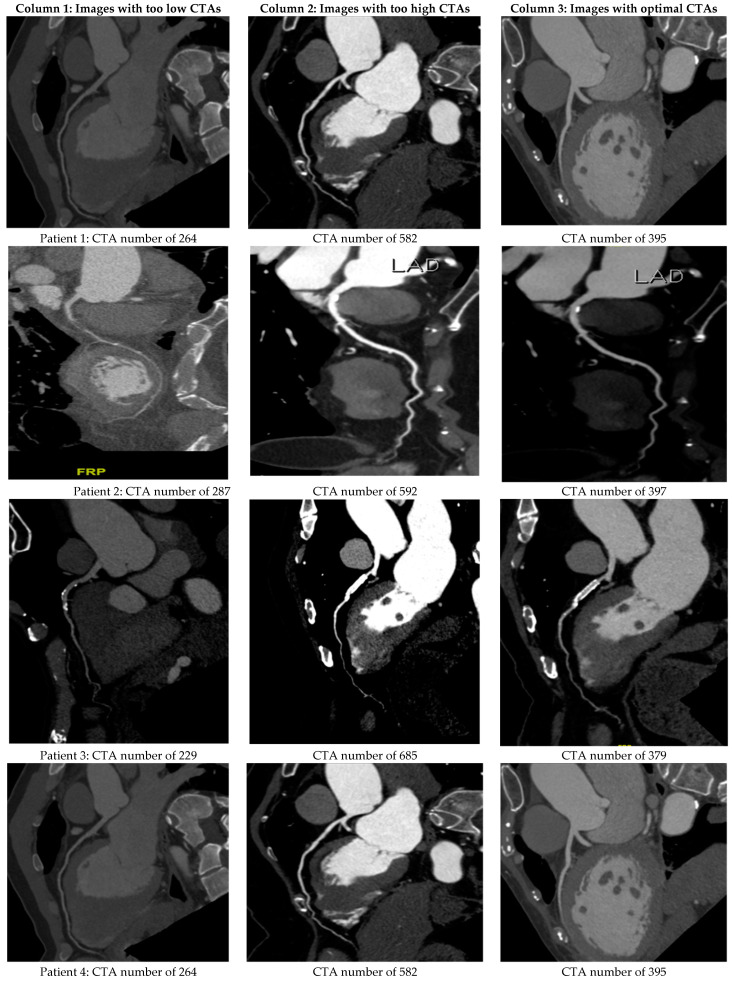
The actual CTA imaging quality for six patients in the testified group of sixty-five. Columns 1, 2, and 3 correspond to the derived CTA numbers that are too low, too high, or optimal, respectively.

**Table 1 diagnostics-13-03354-t001:** The readings of seven factors and actual (original) CT number of the left artery (LA) before the normalization process for 450 patients with cerebral hemorrhage with dizziness, brain aneurysm, stroke, or hemorrhagic stroke diagnosis reported in the Affiliated BenQ Hospital of Nanjing Medical University, Nanjing, Jiangsu, China, from 2020 to 2023.

Risk Factor	Case No. and Variation Range		Parameters of Derived Data		
	Case No./Max.	Case No./Min.	Mean	Median	St. Dev
Age (yr)	162/90	383/24	56	58	13.3
Tube voltage (kVp)	142/140	151/70	100	100	19.0
BSA (m^2^)	142/2.70	135/1.27	1.75	1.73	0.20
HR (#/min)	367/139	170/45	72.1	70.0	12.8
CO (L/min)	142/11.11	367/4.18	6.32	6.10	1.06
CM (c.c.)	89/75.0	268/17.0	36.7	34.5	11.5
DTT (s)	95/5.13	360/2.00	3.08	3.00	0.53
**CT number of LA/400**	**270/2.17**	**170/0.61**	**1.13**	**1.11**	**0.20**

**Table 2 diagnostics-13-03354-t002:** The normalized seven risk factors and actual (original) CT number of LA. Specific patient biological indices following the normal distribution are identified by mean values approaching 0.0.

Risk Factor	Normalized Range		Normalized Data Parameters		
	Case No./Max.	Case No./Min.	Mean	Median	St. Dev
Age (yr)	162/+1	383/−1	−0.02	0.03	0.40
Tube voltage (kVp)	142/+1	151/−1	−0.13	−0.14	0.56
BSA (m^2^)	142/+1	135/−1	−0.33	−0.36	0.28
HR (#/min)	367/+1	170/−1	−0.42	−0.47	0.27
CO (L/min)	142/+1	367/−1	−0.38	−0.45	0.31
CM (c.c.)	89/+1	268/−1	−0.32	−0.40	0.40
DTT (s)	95/+1	360/−1	−0.31	−0.36	0.34
**CT number of LA/400**	**270/+1**	**170/−1**	−0.335	−0.357	1.524

**Table 3 diagnostics-13-03354-t003:** The coefficients of the 29-term semi-empirical formula (cf. Equation (7)) from the calculated outcomes of the *STATISTICA 7.0* program. The factors were all normalized from −1 to +1. Thus, the derived large coefficients significantly dominate the performance of the CTA number of LA prediction.

Biological Index	Factor	Coefficient	After Normalization	
			Value	Rank
AGE	A	*a* _1_	0.165404	24
Tube voltage	B	*a* _2_	−0.626781	**7**
BSA	C	*a* _3_	−0.438880	10
HR	D	*a* _4_	−0.181944	23
CO	E	*a* _5_	0.231918	21
CM	F	*a* _6_	0.348170	17
DTT	G	*a* _7_	−0.377270	14
Age × Tube voltage	A × B	*a* _8_	−0.578918	9
Age × BSA	A × C	*a* _9_	0.131548	25
Age × HR	A × D	*a* _10_	0.255888	19
Age × CO	A × E	*a* _11_	−0.035711	29
Age × CM	A × F	*a* _12_	0.255385	20
Age × DT	A × G	*a* _13_	−0.358334	16
Tube voltage × BSA	B × C	*a* _14_	1.366192	**2**
Tube voltage × HR	B × D	*a* _15_	0.127896	26
Tube voltage × CO	B × E	*a* _116_	−1.453575	**1**
Tube voltage × CM	B × F	*a* _17_	−0.291472	18
Tube voltage × DTT	B × G	*a* _18_	0.198130	22
BSA × HR	C × D	*a* _19_	0.372922	15
BSA × CO	C × E	*a* _20_	0.437779	11
BSA × CM	C × F	*a* _21_	−0.661009	**6**
BSA × DTT	C × G	*a* _22_	0.734438	**4**
HR × CO	D × E	*a* _23_	0.068061	28
HR × CM	D × F	*a* _24_	−0.678175	**5**
HR × DTT	D × G	*a* _25_	0.109320	27
CO × CM	E × F	*a* _26_	0.417461	13
CO × DTT	E × G	*a* _27_	−0.803023	**3**
CM × DTT	F × G	*a* _28_	−0.590657	8
Constant		*a* _29_	−0.419399	29

**Table 4 diagnostics-13-03354-t004:** The readings of seven factors and the actual (original) CT number of the left artery (LA). The verified group of 112 cerebral hemorrhage patients with similar dizziness, brain aneurysm, stroke, or hemorrhagic stroke diagnoses were randomly selected from the original dataset group.

Factor	Range		Derived Data		
	Case No./Max.	Case No./Min.	Mean	Median	St. Dev
Age (yr)	9/88	34/31	55.0	55.0	13.9
Tube voltage (kVp)	3/120	71/100	118.4	120.0	5.5
BSA (m^2^)	68/2.39	25/1.25	1.81	1.81	0.19
HR (#/min)	85/103	36/45	72.3	70.0	12.2
CO (L/min)	68/9.6	25/4.5	6.45	6.42	0.77
CM (c.c.)	19/68	30/32	54.4	56.6	8.9
DTT (s)	81/4.61	102/2.37	3.86	3.88	0.32
**CT number of LA/400**	25/1.58	44/0.88	1.12	1.11	0.14

## Data Availability

Data is unavailable due to privacy or ethical restrictions.

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
