# Peer review of "Inverse Problem Algorithm-Based Time-Resolved Imaging of Head and Neck Computed Tomography Angiography Contrast Kinetics with Clinical Testification"

_diagnostics, 2023, doi:10.3390/diagnostics13213354_

Round 1

Reviewer 1 Report

Comments and Suggestions for Authors

After thoroughly reviewing the submitted manuscript detailing the innovative approach to mitigating the challenges of head and neck CT angiography using IPA-based time-resolved imaging of contrast kinetics, I find the paper very interesting and well-written. The comprehensive study, which incorporated a significant sample size from the Affiliated BenQ Hospital of the Nanjing Medical University, meticulously categorized the patient groups and detailed a novel method for assigning risk factors for cerebral hemorrhage patients.

Utilizing the Inverse Problem Analysis (IPA) technique, resulting in a loss function and variance of 3.1837 and 0.8892, respectively, is commendable. Furthermore, introducing a dimensionless AT to indicate the coincidence between theoretical and practical values adds a notable depth to the study's findings. The meticulous breakdown and computation provided for the first-order nonlinear equation and the method's subsequent practical application in the in-vivo testified group further attest to the study's rigor and relevance. The paper offers valuable insights into CT angiography, showcasing a deep understanding and innovative approach to contrast kinetics in the field. I recommend accepting this paper for publication, as it contributes substantially to our current knowledge.

Comments on the Quality of English Language

Moderate editing of English language required

Author Response

Reply to reviewer #1

After thoroughly reviewing the submitted manuscript detailing the innovative approach to mitigating the challenges of head and neck CT angiography using IPA-based time-resolved imaging of contrast kinetics, I find the paper very interesting and well-written. The comprehensive study, which incorporated a significant sample size from the Affiliated BenQ Hospital of the Nanjing Medical University, meticulously categorized the patient groups and detailed a novel method for assigning risk factors for cerebral hemorrhage patients.

Utilizing the Inverse Problem Analysis (IPA) technique, resulting in a loss function and variance of 3.1837 and 0.8892, respectively, is commendable. Furthermore, introducing a dimensionless AT to indicate the coincidence between theoretical and practical values adds a notable depth to the study's findings. The meticulous breakdown and computation provided for the first-order nonlinear equation and the method's subsequent practical application in the in-vivo testified group further attest to the study's rigor and relevance. The paper offers valuable insights into CT angiography, showcasing a deep understanding and innovative approach to contrast kinetics in the field. I recommend accepting this paper for publication, as it contributes substantially to our current knowledge.

[reply] thank you for the confirmation, to solidify our research result and try to submit to an academic journal is always the first priority for college faculty.

----------------------------------------------------------

Reviewer 2 Report

Comments and Suggestions for Authors

1 Kindly complete the section and contribution of the author.

2 Title of the paper needs significant improvement to make it a research-based topic. 

3 Introduction can be enhanced with clarity.

4 SGD parameters must be include.

5 Novelty of the paper should be mentioned.

6 Recent references have not been included in this paper. 

7 Author should cite some of recent work on his area. 

8 One time a thorough revision is also required to rectify the types.

9 Minor corrections in grammar or spelling mistakes.

10 I strongly suggest a major re-articulation of the entire draft for improving the readability.

11 The authors might think of moving some not-so-essential portions into
  appendixes. Additionally, I find that the initial portion of the paper is making some
  solid claims about the contribution, but I didn't find the same at the end. Also, as
  I mentioned earlier, the lack of clear and coherent articulation is the biggest lacuna
  of this present draft.

12 Equations must be typed in equation editors.

13 Kindly consider the guidelines of journal policy and make it accordingly

14 Do you think the proposed work will work in same domain if they consider in innovation and other fields

15 Innovation strategy must be incorporated as per current 17 SDG goals

Author Response

Reply to reviewer #2

1          Kindly complete the section and contribution of the author.

            [reply] thank you for your reviewing.

2          Title of the paper needs significant improvement to make it a research-based topic.

            [reply] The new title is “Inverse Problem Algorithm-based Time-Resolved Imaging of Head and Neck CTA Contrast Kinetics with Clinical Testification“ to coincide the special section “Head and Neck CT Angiography Challenge “ of this journal

3          Introduction can be enhanced with clarity.

            [reply] We did our best to improve the introduction.

4          SGD parameters must be included.

            [reply] Instead of the Stochastic Gradient Descent (SGD) in the STATISTICA 7.0 program, we assessed the loss function (3.184), variance (0.89), and regression correlation coefficient (0.943). Besides, we moved some less important details on STATISTICA to Appendix, according to your suggestion in comment 11.

5          Novelty of the paper should be mentioned.

            [reply] Thank you for finding this omission. We revised the introduction accordingly.

6          Recent references have not been included in this paper.

            [reply] While most cited references were published after 2012, we added most recent two [12] of 2023. And [17] of 2017.

7          Author should cite some of recent work on his area.

            [reply] thank you for the reminding. Most of the quoted references are correlated to linear regression, optimization or specially in IPA algorithm.

8          One time a thorough revision is also required to rectify the types.

            [reply] thank you for the reminding. The revised manuscript was polished by professional English editor from Springer Nature with Ph.D. degree.

9          Minor corrections in grammar or spelling mistakes.

            [reply] Sorry for overlooking these lapses. The revised paper was also double-checked with AJE digital editing tool and Grammarly Premium

10        I strongly suggest a major re-articulation of the entire draft for improving the readability.

            [reply] We did our best to follow your recommendations. The manuscript’s revisions are over 20%.

11        The authors might think of moving some not-so-essential portions into appendixes. Additionally, I find that the initial portion of the paper is making some solid claims about the contribution, but I didn't find the same at the end. Also, as I mentioned earlier, the lack of clear and coherent articulation is the biggest lacuna of this present draft.

            [reply] Thank you for this idea. We moved STATISTICA details and Figure 1 to the Appendix. We also revised the introduction and conclusion to provide more clear vision of the application of IPA in handling the medical issue.

12        Equations must be typed in equation editors.

            [reply] thank you for the reminding. We revise as recommended.

13        Kindly consider the guidelines of journal policy and make it accordingly

            [reply] thank you for the reminding. We revise as recommended

14        Do you think the proposed work will work in same domain if they consider in innovation and other fields

            [reply] The IPA algorithm has been successfully applied to other topics of medical field, e.g. in works [9] and [11] published by our research team.

15        Innovation strategy must be incorporated as per current 17 SDG goals Comments on the Quality of English Language

            [reply] Unfortunately, the 17 SDG goals regulations are too general to be directly applied to this specific topic. As to health direction of UN’s SDG, our innovation strategy is straightforward – optimizing clinical examination parameters and reducing time and amount of contrast media injected in patients. As to the Quality of English Language, the available online tools (American Journal Editor grammar check and Grammarly Premium) ranked the quality of English language of the revised paper as 8.4 and 8.8 of 10, respectively.

Many thanks for your comments and suggestions to improve our paper.

----------------------------------------------------------

Reviewer 3 Report

Comments and Suggestions for Authors

The authors used Inverse Problem Algorithm (IPA)-based time resolved imaging of contrast kinetics to solve the challenge of head and neck CT angiography. 627 cerebral hemorrhage patients with the dizziness, brain aneurysm, stroke, or hemorrhagic stroke diagnosis were randomly categorized into the original dataset, verification group, and in-vivo testified group. The authors used STATISTICA 7.0 to obtain the 29 items of first-order nonlinear equation via the Inverse Problem Analysis (IPA) technique. Then the optimal amount of contrast media for approaching the CTA number 400 of LA was found. The result was verified on a dataset with 65 patients.

Major comments:

1. The last paragraph of Section 1. Introduction should be rewritten. A brief introduction of the following sections and the major contributions of the manuscript should be given.

2. It’s not clearly described why 29-term coefficients are used.

3. The tool used in the study: STATISTICA 7.0 is not mentioned in the keywords.

Minor comments:

1. Equ. 7 might exceed the range of the page.

2. Please keep the font of the variables in the paragraphs and equations consistent through the manuscript.

3. Equ. 2, the values of n and m are not given.

4. Caption of Figure 1 is not accurate. It’s not the ‘performance’ of STATISTICA 7.0.

Author Response

Reply to reviewer #3

The authors used Inverse Problem Algorithm (IPA)-based time resolved imaging of contrast kinetics to solve the challenge of head and neck CT angiography. 627 cerebral hemorrhage patients with the dizziness, brain aneurysm, stroke, or hemorrhagic stroke diagnosis were randomly categorized into the original dataset, verification group, and in-vivo testified group. The authors used STATISTICA 7.0 to obtain the 29 items of first-order nonlinear equation via the Inverse Problem Analysis (IPA) technique. Then the optimal amount of contrast media for approaching the CTA number 400 of LA was found. The result was verified on a dataset with 65 patients.

Major comments:

 The last paragraph of Section 1. Introduction should be rewritten. A brief introduction of the following sections and the major contributions of the manuscript should be given.

[reply] thank you for the reminding. We revise the last section of introduction as recommended.

  1. It’s not clearly described why 29-term coefficients are used.

[reply] thank you for the reminding. The description of specific 29-term coefficient of semi-empirical formula was listed in sec 2.1.2, L107-111.

  1. The tool used in the study: STATISTICA 7.0 is not mentioned in the keywords.

[reply] thank you for the reminding. We add in the list of keywords

Minor comments:

 Eq. 7 might exceed the range of the page.

[reply] thank you for the reminding. The format editor can handle the format

  1. Please keep the font of the variables in the paragraphs and equations consistent through the manuscript.

[reply] thank you for the reminding. We revise in the text.

  1. Eq. 2, the values of n and m are not given.

[reply] thank you for the reminding. n is the number of patients, it equals 450, or 112 depended on the database. m is the number of coefficient and equals 29 in this study.

  1. Caption of Figure 1 is not accurate. It’s not the ‘performance’ of STATISTICA 7.0.

[reply] thank you for the reminding. We revise as “STATISTICA 7.0 worksheet”.

Many thanks for your comments and suggestions to improve our paper.

----------------------------------------------------------

Reviewer 4 Report

Comments and Suggestions for Authors

General comment

The manuscript has potential, but it requires significant revisions to improve clarity, organization, and readability. Simplify the methodology section, restructure the results section, and ensure that the discussion emphasizes the clinical relevance of the findings. Additionally, revise the title and abstract to make them more concise and descriptive.

Major comments

1.       The manuscript lacks clarity in several areas, particularly in the presentation of the methodology and results. The writing could benefit from restructuring and more concise explanations to improve readability and understanding.

2.       The title is long and somewhat unclear. Consider revising it to provide a more concise and descriptive title that reflects the research focus. The abstract should be rewritten to provide a more succinct summary of the research objectives, methods, and key findings.

3.       The introduction provides some background on the research topic, but it could be more concise. Consider focusing on the specific research question and objectives to engage the reader more effectively.

4.       The methodology section is overly detailed and lacks clarity. The description of the Inverse Problem Algorithm (IPA) technique and its application in this study needs to be simplified and explained step by step. It is also important to clarify how the risk factors were selected and why they are relevant.

5.       The results section is dense and challenging to follow. Consider presenting the results in a more organized manner, using tables and figures to illustrate key findings. Additionally, provide a clear interpretation of the results and their implications for the research question.

6.       The discussion section should focus on interpreting the results in the context of the research question and objectives. It is essential to discuss the clinical significance of the findings and how they contribute to the field of head and neck CT angiography.

Minor comments

1.       Check for typographical errors and grammatical issues, particularly in the Methods and Results sections.

2.       Provide a more concise and focused conclusion that summarizes the key findings and their implications.

3.       Consider shortening some of the overly long sentences and paragraphs for improved readability.

Comments on the Quality of English Language

Moderate editing of English language required.

Author Response

Reply to reviewer #4

General comment

 The manuscript has potential, but it requires significant revisions to improve clarity, organization, and readability. Simplify the methodology section, restructure the results section, and ensure that the discussion emphasizes the clinical relevance of the findings. Additionally, revise the title and abstract to make them more concise and descriptive.

[reply] thank you for the reminding. We have thoroughly revised the text by professional English editor to smooth the flow.

Major comments

  1. The manuscript lacks clarity in several areas, particularly in the presentation of the methodology and results. The writing could benefit from restructuring and more concise explanations to improve readability and understanding.

[reply] thank you for the reminding. We revise the introduction and conclusion to add the aim and future application of IPA technique in medical field.

  1. The title is long and somewhat unclear. Consider revising it to provide a more concise and descriptive title that reflects the research focus. The abstract should be rewritten to provide a more succinct summary of the research objectives, methods, and key findings.

[reply] thank you for the reminding. We revise as “Inverse Problem Algorithm-based Time-Resolved Imaging of Head and Neck CTA Contrast Kinetics with Clinical Testification” for more research-based sound.

  1. The introduction provides some background on the research topic, but it could be more concise. Consider focusing on the specific research question and objectives to engage the reader more effectively.

[reply] thank you for the reminding. We revise the final part of introduction to emphasize the potential application of IPA in medical field, and the others remains to include background review and rational study to fulfill other reviewers’ comments.

  1. The methodology section is overly detailed and lacks clarity. The description of the Inverse Problem Algorithm (IPA) technique and its application in this study needs to be simplified and explained step by step. It is also important to clarify how the risk factors were selected and why they are relevant.

[reply] thank you for the reminding. The IPA algorithm is a newly developed technique to predict the possible trends of interested syndrome, therefore, some technical term in operating the STATISTICA 7.0 needs to be further addressed in the text. We also move the description of IPA to Appendix for more concise reading.

  1. The results section is dense and challenging to follow. Consider presenting the results in a more organized manner, using tables and figures to illustrate key findings. Additionally, provide a clear interpretation of the results and their implications for the research question.

[reply] thank you for the reminding. We revise the section head of result for readers to follow and still hold the part of sec 4.1 in chapter of Discussion for further elaboration.

  1. The discussion section should focus on interpreting the results in the context of the research question and objectives. It is essential to discuss the clinical significance of the findings and how they contribute to the field of head and neck CT angiography.

[reply] thank you for the reminding. Sec 4.2 is specifically defined as the interpretation of derived coefficients for semi-empirical formula, L255-270

Minor comments

 Check for typographical errors and grammatical issues, particularly in the Methods and Results sections.

[reply] thank you for the reminding. We revise thoroughly according to your recommendation.

  1. Provide a more concise and focused conclusion that summarizes the key findings and their implications.

[reply] thank you for the reminding. We revise the conclusion to fulfill you and other reviewer’s suggestions too.

  1. Consider shortening some of the overly long sentences and paragraphs for improved readability.

[reply] thank you for the reminding. We revise thoroughly the text by professional English editor.

Many thanks for your comments and suggestions to improve our paper.

======================================================

Round 2

Reviewer 3 Report

Comments and Suggestions for Authors

The authors used Inverse Problem Algorithm (IPA)-based time resolved imaging of contrast kinetics to solve the challenge of head and neck CT angiography. 627 cerebral hemorrhage patients with the dizziness, brain aneurysm, stroke, or hemorrhagic stroke diagnosis were randomly categorized into the original dataset, verification group, and in-vivo testified group. The authors used STATISTICA 7.0 to obtain the 29-term first-order nonlinear equation via the Inverse Problem Analysis (IPA) technique. Then the optimal amount of contrast media for approaching the CTA number 400 of LA was found. The result was verified on a dataset with 65 patients.

----------------------------------------------------------

Reviewer 4 Report

Comments and Suggestions for Authors

All the major and minor concerns have been addressed.